# Development and validation of a set of novel and robust 4-lncRNA-based nomogram predicting prostate cancer survival by bioinformatics analysis

**Peng Zhang[1], Xiaodong Tan[2], Daoqiang Zhang[3], Qi Gong[3], Xuefeng Zhang[1]***

**1** Department of Urology, Weihai Central Hospital, Weihai, Shandong, China, **2** Clinical Lab, Weihai Central Hospital, Weihai, Shandong, China, **3** Weihai Key Laboratory of Autoimmunity, Weihai Central Hospital, Weihai, Shandong, China

* xuefengzhangdoctor@hotmail.com

## Abstract

### Background and objective

Accumulating evidence shows that long noncoding RNAs (lncRNAs) possess great potential in the diagnosis and prognosis of prostate cancer (PCa). Therefore, this study aimed to construct an lncRNA-based signature to more accurately predict the prognosis of different PCa patients, so as to improve patient management and prognosis.

### Methods

Through univariate and multivariate Cox regression analysis, this study constructed a 4 lncRNAs-based prognosis nomogram for the classification and prediction of survival risk in patients with PCa based on TCGA data. Then we used the data of TCGA and ICGC to verify the performance of our prediction model. The receiver operating characteristic curve was plotted for detecting and validating our prediction model sensitivity and specificity. In addition, Cox regression analysis was conducted to examine whether the signature's prediction ability was independent of additional clinicopathological variables. Possible biological functions for those prognostic lncRNAs were predicted on those 4 protein-coding genes (PCGs) related to lncRNAs.

### Results

Four lncRNAs (HOXB-AS3, YEATS2-AS1, LINC01679, PRRT3-AS1) were extracted after COX regression analysis for classifying patients into high and low-risk groups by different OS rates. As suggested by ROC analysis, our proposed model showed high sensitivity and specificity. Independent prognostic capability of the model from other clinicopathological factors was indicated through further analysis. Based on functional enrichment, those action sites for prognostic lncRNAs were mostly located in the extracellular matrix and cell membrane, and their functions are mainly associated with the adhesion, activation and transport of the components across the extracellular matrix or cell membrane.

**Data Availability Statement:** The data used in this study are third party data from https://portal.gdc.cancer.gov/ (TCGA) and https://dcc.icgc.org/

releases/current/Projects/PRAD-CA (ICGC) and can be accessed following the protocol outlined in the Methods section and Supporting Information files.

**Funding:** The present study was funded by grants from Weihai Key Laboratory of Autoimmunity, Weihai Central Hospital (grant number: 2017GGH11).

**Competing interests:** The authors have declared that no competing interests exist.

**Abbreviations:** PCa, Prostate cancer; lncRNAs, Long noncoding RNAs; TCGA, The Cancer Genome Atlas; DElncRNAs, Differently expressed lncRNAs; ICGC, International Cancer Genome Consortium; ROC, Receiver operating characteristic analysis; GO, Gene ontology; KEGG, Kyoto Encyclopedia of Genesand Genomes; PCG, Protein-coding gene; HR, Hazard ratio; OS, Overall survival; AUC, Area under curve; GDC, Genomic Data Commons; AIC, Akaike Information Criterion; C-index, Concordance index; SRS, Survival risk score; CI, Confidence interval; ECM, Extracellular matrix; CAMs, Cell adhesion molecules.

## Conclusion

Our current study successfully identifies a novel candidate, which can provide more convincing evidence for prognosis in addition to the traditional clinicopathological indicators to predict the PCa survival, and laying the foundation for offering potentially novel therapeutic treatment. Additionally, this study sheds more lights on the PCa-related molecular mechanisms.

## Introduction

Prostate cancer (PCa) represents a frequently occurring cancer in men globally [1]. A number of clinicopathological factors, such as the Gleason score, margin status, or the TNM stage, are incorporated into prior models to diagnose and detect PCa after treatment [2]. Of such factors, Gleason score exhibits the highest sensitivity and effectiveness. Nonetheless, the subjectivity and sampling error related to the assessment of Gleason score are the prominent confounders. Recently, increasing research attempts to establish the gene molecular signature for enhancing the prediction ability for PCa [3]. Nonetheless, little existing research has examined the possible roles of lncRNAs in the prediction of PCa survival risk as the new models. lncRNAs have been usually referred to as the RNA transcripts with the length of over 200 nucleotides (nt) that cannot encode proteins [4]. It is increasingly suggested that, lncRNAs have exerted vital parts in numerous biological processes through regulating gene expression, or proliferation and apoptosis of cells [5]. The aberrant lncRNAs are related to tumorigenesis, which may serve as the oncogenes or tumor suppressor genes. In addition to their parts in tumor genesis and development, lncRNAs may serve as the candidate biomarkers [6]. So far, bioinformatic analysis is substantially used for molecular biological experiments or in clinic. Therefore, this work aimed to employ bioinformatic analysis to discover those differently expressed lncRNAs (DElncRNAs) between PCa and non-carcinoma tissue samples from the sequencing data obtained through The Cancer Genome Atlas (TCGA). Then, Cox regression analysis together with the survival associated risk score formula was used to develop a nomogram for prognosis based on four lncRNAs, so as to differentiate cases with favorable or dismal prognostic outcome. Then we used TCGA and International Cancer Genome Consortium (ICGC) datasets to verify the performance of our prediction model. The receiver operating characteristic (ROC) curve was also plotted, with a higher area under curve (AUC) indicated favorable model sensitivity and specificity. In addition, univariate as well as multivariate Cox regression suggested that this4 lncRNAs-based prognosis nomogram showed higher independence and robustness in predicting prognosis compared with additional clinicopathological variables. Besides, analyzing the functions and involved pathways for these 4 lncRNAs shed more lights on the lncRNAs prediction ability and the possible molecular mechanism.

## Materials and methods

### The overall analysis route in this work

The general research analysis route can be observed from **Fig 1**. First, those transcriptome RNA-seq expression profiles from 551 cases were obtained based on the TCGA database. Then lncRNA was analyzed for differential expression and batch survival. Univariate and multivariate COX regression analysis was performed for differentially expressed lncRNAs to establish a gene model of 4-lncRNA. Then we combined ICGC database data to verify the prediction

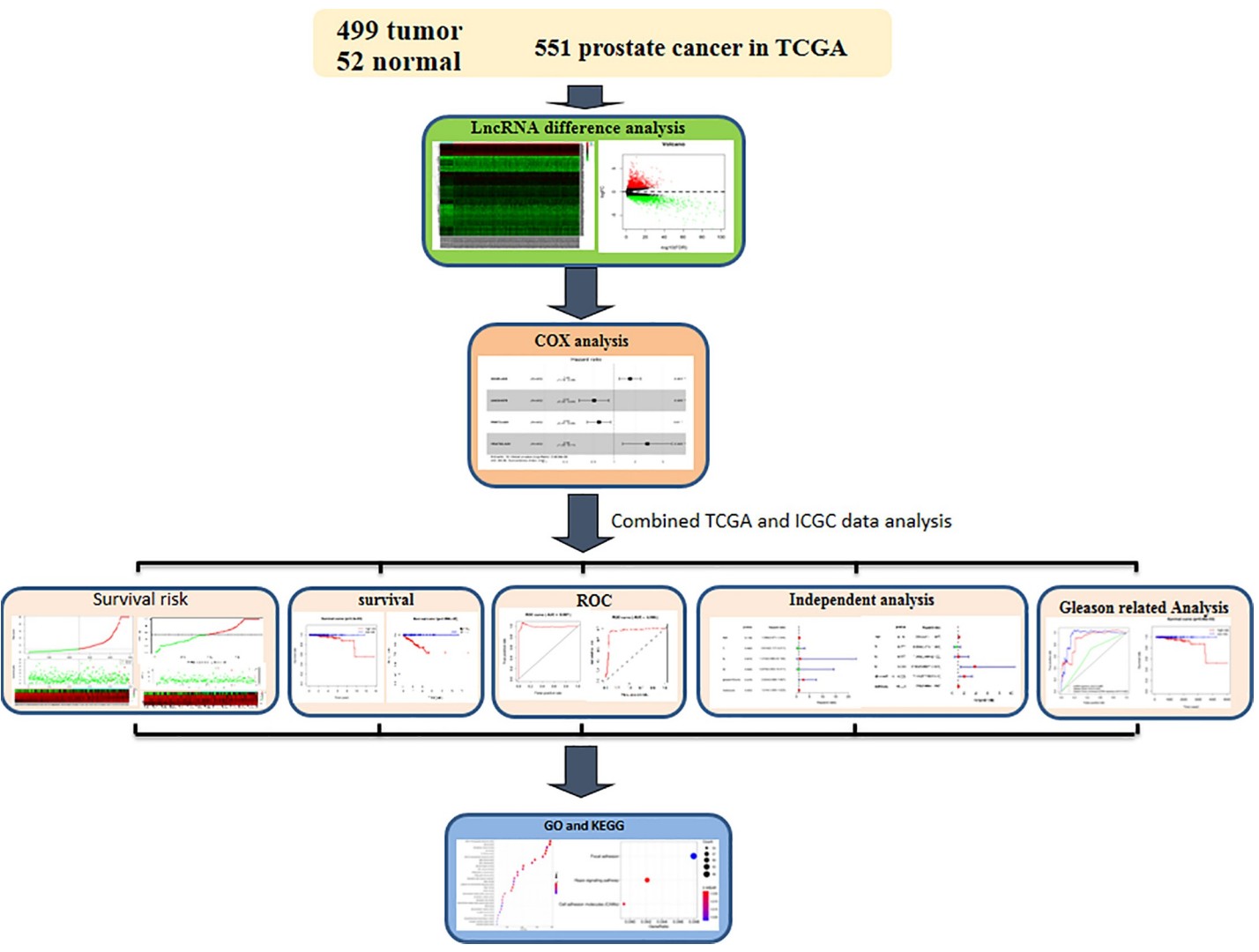

**Fig 1. The overall analysis design of our study.**

performance of the prediction model. Survival analysis, ROC curve analysis, and patient risk heat plot, risk curve and survival status plot were performed respectively for our model. Target genes of lncRNAs were predicted by co-expression, and then the target genes were analyzed by functional clustering analysis. Finally, independent prognostic analysis and correlation analysis were conducted between our 4-lncrNA nomogram and other common clinical characteristics, as well as further stratified analysis and combined analysis of Gleason Score.

## Datasets

LncRNA RNA-seq data (HTSeq-Counts) in the United States and China including PCa and control samples and corresponding clinical data of PCa patients were extracted based on the publicly available Genomic Data Commons (GDC) data portal (https://portal.gdc.cancer.gov/) and TCGA data portal (https://dcc.icgc.org/releases/ current/Projects/PRAD-CA) respectively. Altogether 551 gene expression profiling samples were collected from TCGA, comprising 499 PCa and 52 non-carcinoma tissues. 435 gene expression profiling samples were took from

ICGC, comprising 393 PCa and 42 non-carcinoma tissues. Samples with insufficient clinical data or those with OS < 3 months were eliminated from the present research.

## Identification of DElncRNAs between PCa and non-carcinoma tissue samples

In line with our inclusion criteria, clinicopathological data from 397 PCa cases were obtained from TCGA database (**Table 1**), and clinical data from 378 PCa cases were obtained from ICGC database (**Table 2**). Clinical covariates for both cases and cancers can be observed from **Tables 1 and 2**. The R4.0.1 software was employed for analysis. For identifying the candidate lncRNAs in later survival analysis, the trimmed mean of M values was adopted to normalize and differentially analyze the expression profiles by edgeR package from Bioconductor [7]. The heterogeneities of lncRNA expression in PCa compared with adjacent normal tissues were presented in the form of log2FC together with related P-value. In this study, $|log2FC| > 1$ and false discovery rate (FDR) q < 0.05 were selected as the thresholds. Thereafter, the R package pheatmap function (version 1.0.12) was used to perform unsupervised hierarchical clustering according to DElncRNA expression levels.

## Survival analysis and development of the lncRNAs expression model

Firstly, RNA-seq expression data were normalized according to log2. Later, the Survival R package from CRAN (https://rweb.stat.umn.edu/R/site-library/survival/html/00Index.html) was used to assess the relationships of DElncRNAs with patient OS through univariate Cox proportional hazards regression. lncRNAs with P-value < 0.05 were identified to be the potential variables, which were incorporated into the stepwise multivariate Cox regression examined through Akaike Information Criterion (AIC), which help to determine the optimal exchange between model complexity and the optimal informative and explanatory effectiveness.

**Table 1. PCa patients from TCGA database clinicopathological features.**

| Features | | Patients (N = 397) | |
|---|---|---|---|
| | | n | % |
| Age(years) | < 60 | 153 | 38.54 |
| | ≥ 60 | 244 | 61.46 |
| Gleason score | ≤ 7 | 215 | 54.16 |
| | ≥ 8 | 182 | 45.84 |
| T stage | T2 | 130 | 32.75 |
| | T3-4 | 267 | 67.25 |
| N stage | N0 | 322 | 81.11 |
| | N1 | 75 | 18.89 |
| M stage | M0 | 311 | 78.34 |
| | M1 | 86 | 21.66 |
| Race | American indian or alaska native | 1 | 0.25 |
| | Asian | 10 | 2.52 |
| | Black or african american | 46 | 11.59 |
| | White | 329 | 82.87 |
| | Not reported | 11 | 2.77 |
| Vital status | Alive | 388 | 97.73 |
| | Dead | 9 | 2.27 |

**Table 2. PCa patients from ICGC database clinicopathological features.**

| Features | | Patients (N = 378) | |
|---|---|---|---|
| | | n | % |
| Age(years) | < 60 | 179 | 47.35 |
| | ≥ 60 | 199 | 52.65 |
| Gleason score | ≤ 7 | 144 | 38.10 |
| | ≥ 8 | 234 | 61.90 |
| T stage | T2 | 74 | 19.58 |
| | T3-4 | 304 | 80.42 |
| N stage | N0 | 280 | 74.07 |
| | N1 | 98 | 25.93 |
| M stage | M0 | 252 | 66.67 |
| | M1 | 126 | 33.33 |
| Vital status | Alive | 344 | 91.01 |
| | Dead | 34 | 8.99 |

## Risk stratification, survival curve and concordance index (C-index)

The riskscore values for all patients were determined by the following formula (Risk score = $0.524052278 \times$ HOXB-AS3 $- 0.663887578 \times$ LINC01679 $- 0.504710478 \times$ PRRT3-AS1 $+ 1.092940489 \times$ YEATS2-AS1) on the basis of multivariate Cox regression. Using the as-prepared risk scoring system, all cases were classified as high- or low-risk group by median riskscore value. Thereafter, heterogeneities of OS time between these two groups were examined through two-sided log-rank test. The Kaplan-Meier (K-M) method was employed to plot the OS curves for these two groups. Besides, the AUC values were determined for comparing the model sensitivity and specificity in predicting OS by the "survival ROC" of R package. The model discrimination ability was evaluated by calculating the C-index.

## Predictive independence of our 4 lncRNAs-based nomogram for survival rate from additional clinicopathological factors

The independence of our 4 lncRNAs-based nomogram from additional clinicopathological factors (such as age, Gleason score, and TNM classification) in predicting OS was examined by univariate as well as multivariate Cox regression. OS was used to be a dependent variable, whereas the constructed lncRNA nomogram together with additional common clinical factors were used to be the independent variables.

## Joint analysis of the four-lncRNA signature with Gleason score

In order to test whether our model could accurately predict the prognosis of patients with Gleason score ≥ 8, a stratified analysis was carried out. To verify whether our lncRNA model could enhance Gleason score's accuracy in PCa survival risk prediction, the ROC curves of the three comparisons was drawn and the AUCs were calculated.

## Gene Ontology (GO) and Kyoto Encyclopedia of Genes and Genomes (KEGG) pathway analyses

Using the z-test and two-sided Pearson correlation coefficients, Pearson correlation coefficients were examined between those 4 lncRNAs expression and PCGs, so as to discover the possible biological processes together with related pathways involving those predictive

lncRNAs. Typically, PCGs showing |Pearson correlation coefficient| > 0.40 and P < 0.01 were identified to show positive or negative relationship with lncRNAs. GO and KEGG analyses were conducted on the lncRNA-related PCGs via the Database for Annotation, Visualization, and Integrated Discovery at the threshold of false discovery rate (FDR) q < 0.05.

## Results

### Identification of DElncRNAs between PCa and non-carcinoma tissue samples

Based on thresholds of |log2FC| > 1.0 and false discovery rate (FDR) q < 0.05, altogether 451 lncRNAs, among which, 307 were up-regulated whereas 144 were down-regulated, were discovered to show different expression between PCa and the non-carcinoma tissue samples, and they were applied in later stepwise survival analysis (**S1 Table**). The expression profiles were intuitively reflected by volcano plots (**Fig 2**). According to the DElncRNAs profiles, the unsupervised hierarchical cluster analysis was carried out, which suggested the possibility to distinguish between PCa and non-carcinoma samples (**S1 Fig**). To identify prognosis-related lncRNAs which are associated with patients' OS in PCa, the lncRNA expression profiles were evaluated by univariate Cox regression analysis data. 36 lncRNAs related to OS were selected at the threshold of 0.05 (**S2 Table**), which were then applied for stepwise multivariate Cox regression analysis, and four lncRNAs therein (HOXB-AS3, LINC01679, PRRT3-AS1,

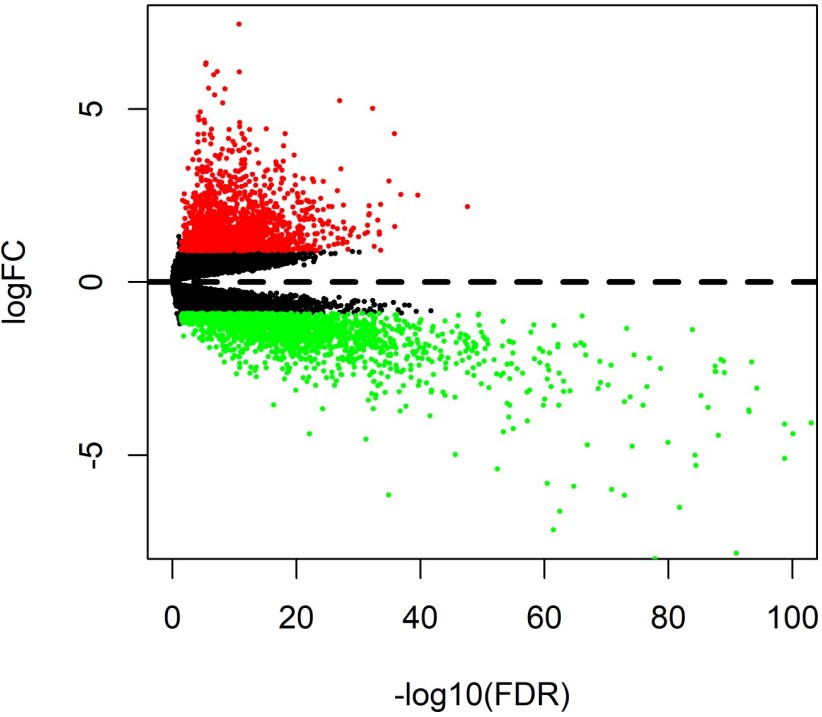

**Fig 2. DElncRNAs between prostate cancers and non-carcinoma tissue samples.** Red dots represent differentially upregulated lncRNAs, green dots represent differentially downregulated lncRNAs and then dark dots represent no differentially expressed genes.

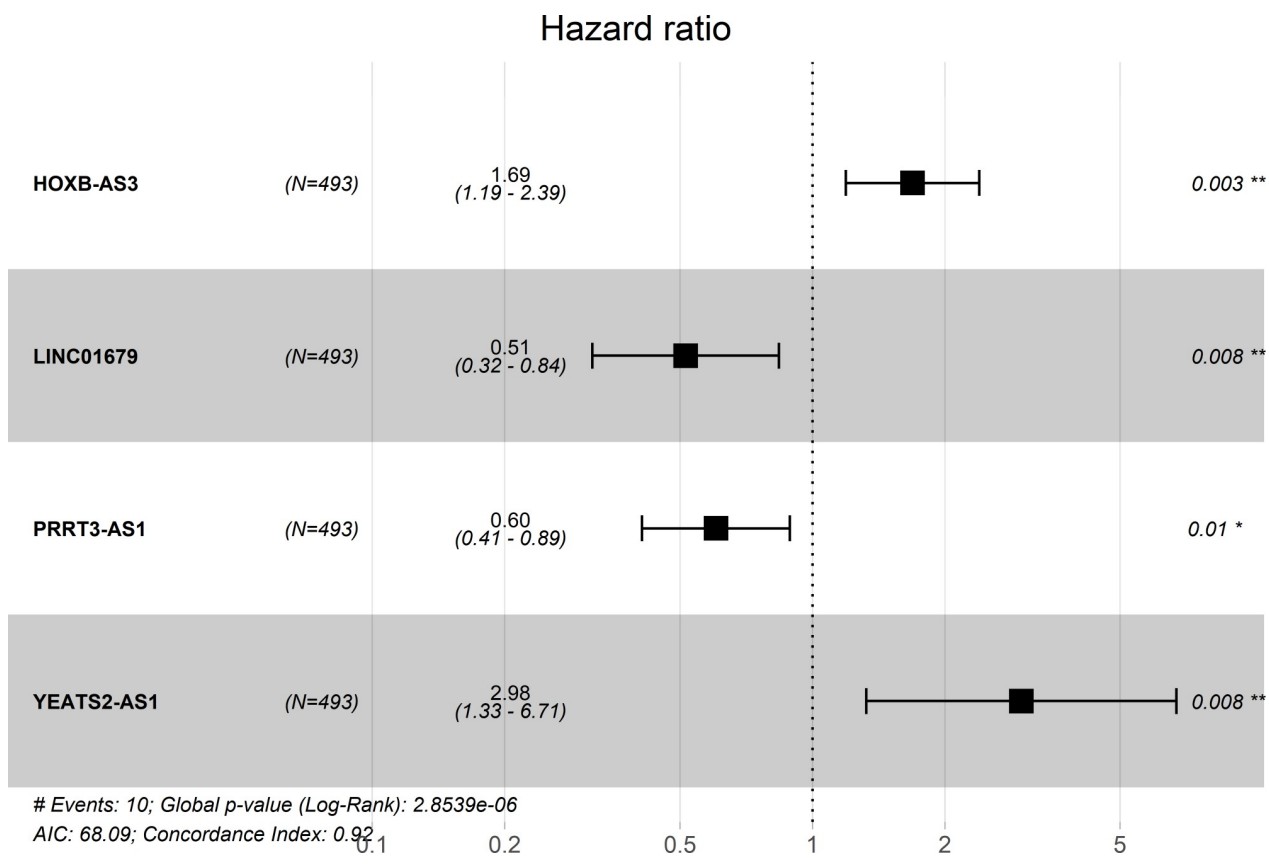

**Fig 3. Four lncRNAs Forest map formed by multivariate Cox regression analysis.**

YEATS2-AS1) (as shown in **Fig 3**, **Table 3**) were ultimately screened out from the 451 lncRNAs identified before to establish a predictive model.

Expression profiles of those 4 lncRNAs were integrated with the related regression coefficients to construct a prognosis nomogram. Based on the above-mentioned analysis, the risk-score formula was the sum of 4 lncRNAs expression levels weighted by the corresponding relative regression coefficients obtained upon multivariate Cox regression, as shown below: survival risk score = (0.524052278 × HOXB-AS3 expression) + (1.092940489 × YEATS2-AS1 expression) + (-0.663887578 × LINC01679 expression) + (-0.504710478 × PRRT3-AS1 expression). Of these four lncRNAs, two had positive coefficients upon multivariate Cox regression analysis, associated with high risk because the up-regulated level indicated the reduced patient

**Table 3. Overview for the four prognostic lncRNAs related to PCa patient OS.**

| gene | coef[a] | exp(coef)[b] | se(coef)[c] | z | Pr(> |z|) |
|---|---|---|---|---|---|
| HOXB-AS3 | 0.524052278 | 1.688857522 | 0.178121024 | 2.942113552 | 0.003259804 |
| LINC01679 | -0.663887578 | 0.514845935 | 0.249133131 | -2.664790406 | 0.007703632 |
| PRRT3-AS1 | -0.504710478 | 0.603680329 | 0.197208039 | -2.559279435 | 0.010488939 |
| YEATS2-AS1 | 1.092940489 | 2.983032765 | 0.413609893 | 2.642442815 | 0.008231036 |

[a]Coef: coefficient

[b]exp(coef): hazard ratio

[c]se(coef): the range of values at hazard ratio.

OS (HOXB-AS3 and YEATS2-AS1, Coef > 0) and the remaining two lncRNAs (LINC01679 and PRRT3-AS1, Coef < 0) were shown negative coefficients upon Cox regression analysis, which indicated that such lncRNAs were protective, because cases having up-regulate lncRNAs levels tended to show extended OS relative to patients having decreased expression (**Fig 3**).

## Favorable performance for the 4-lncRNA prognostic model in the prediction of OS for PCa cases

The 493 cases were classified as high- (n = 246) or low-risk (n = 247) group in line with the median riskscore (0.943) obtained based on those 4 lncRNAs expression levels (also defined as the survival risk score, SRS) in the TCGA database (**Fig 4** and **S3 Table**). According to the expression levels of these 4 lncRNAs in the ICGC database, the median SRS was 1.561, and 392 patients were divided into the high (n = 196) group and the low (n = 196) group (**Fig 5 and S4 Table**). Difference in survival was determined by log-rank test. The K-M method was used for survival analysis. It was illustrated from **Fig 6** (TCGC) and **Fig 7** (ICGC) that, the KM OS curves for both groups on the basis of 4 lncRNAs showed notable difference (p = 3.3e-03 and 3.06E-09). Typically, the low-risk group showed significant correlation with favorable prognosis compared with high-risk group. The AUC values of time-dependent ROC curves were calculated to evaluate our constructed nomogram prognostic ability. The greater AUC value is indicative of the higher nomogram performance, and that AUC of >0.90 has excellent performance. Based on the analysis results of ROC curves, the AUC values at 1, 3 and 5 years of TCGA database were 0.997, 0.929 and 0.928, separately (**Fig 8**), and the 1-year, 3-year and 5-year AUC of ICGC database were 0.946, 0.928 and 0.905 respectively (**Fig 9**), revealing the high sensitivity and specificity of our 4 lncRNAs-based signature in the prediction of OS risk for PCa cases. Additionally, the model C-index was calculated, (TCGA C-index = 0.9203, CI: 0.8482–0.9924, p-value = 3.13447e-30); ICGC C-index = 0.9636, CI: 0.9512–0.9759, p-value = 0, exhibiting good model performance.

## lncRNAs-based nomogram's predictive ability was not dependent on other clinicopathological factors

For investigating the distinguishing ability of our constructed 4 lncRNAs-based signature of the survival risk for PCa cases after considering additional possible traditional prognostic factors, univariate as well as multivariate analysis was conducted for evaluating the model independent prognostic significance. The multivariate analysis results demonstrated that our 4 lncRNAs-based signature might be used to be the potent factor to independently predict PCa OS rate from other clinical factors (TCGA hazard ratio (HR) = 1.014, 95% CI 1.005–1.023, P = 0.003; ICGC HR = 1.011, 95% CI 1.006–1.016, P < 0.001), shown in **Tables 4, 5, Figs 10 and 11**, compared with conventional clinicopathological factors such as age, Gleason score and TNM classification.

## Contrast of the four-lncRNA signature with Gleason score

Hierarchical analysis showed that PCa with Gleason score > 7 could be further stratified as 2 groups that had different survival by the nomogram (TCGA log-rank test, P = 2.359e-02, **Fig 12**; ICGC log-rank test, P = 3.847e-06, **Fig 13**). Combining the 4 lncRNAs-based signature and the Gleason score remarkably increased the model prognostic capacity compared with the Gleason score alone (TCGA AUC, 0.949 vs. 0.896 vs. 0.633, **Fig 14**; ICGC AUC, 0.941 vs. 0.909 vs. 0.808, **Fig 15**).

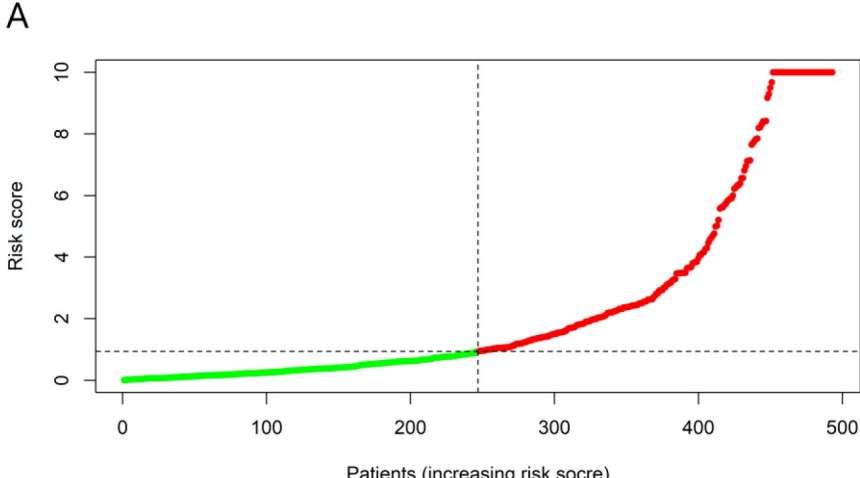

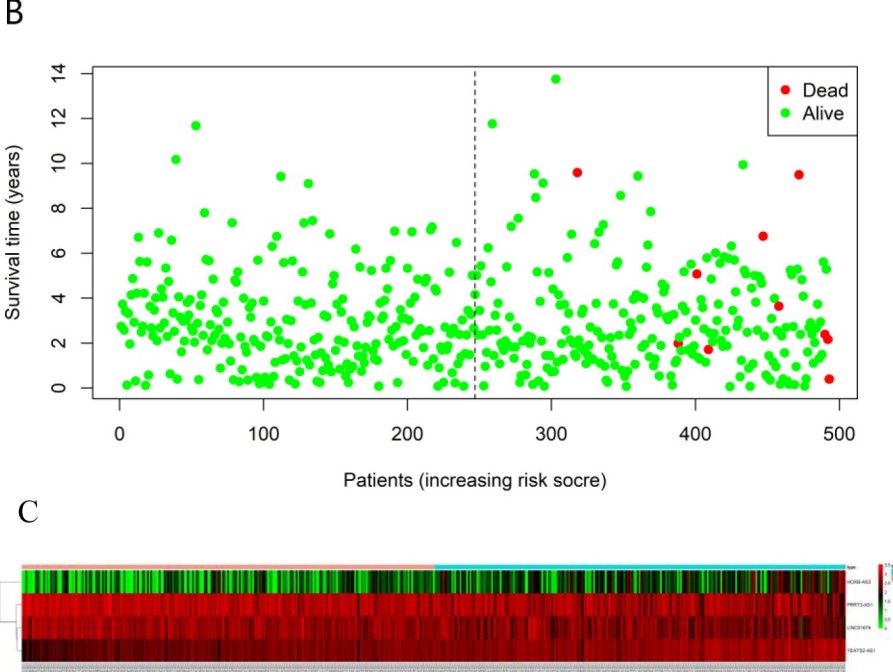

**Fig 4. Ability of the 4 lncRNAs-based signature in predicting the prognosis for PCa cases from TCGA database.**
(A) Risk score distribution for patients. (B) PCa patient survival time. (C) Expression heat map for those four lncRNAs incorporated into the prognosis model. The vertical black dotted line stands for the optimum threshold to divide cases to high- or low-risk group.

## Identification of the prognostic lncRNAs signature associated biological functional characteristic

After determining the associations between those 4 lncRNAs and the PCGs, this study selected co-expression between 1 of those 4 lncRNAs and 4080 PCGs (|Pearson correlation coefficient| > 0.4, P-value < 0.05, and q-value < 0.05). Afterwards, GO and KEGG analyses were conducted for the lncRNAs-related PCGs to display the possible functions for those 4 prognostic lncRNAs. The 4080 PCGs were most significantly enriched into 28 GO terms and they were

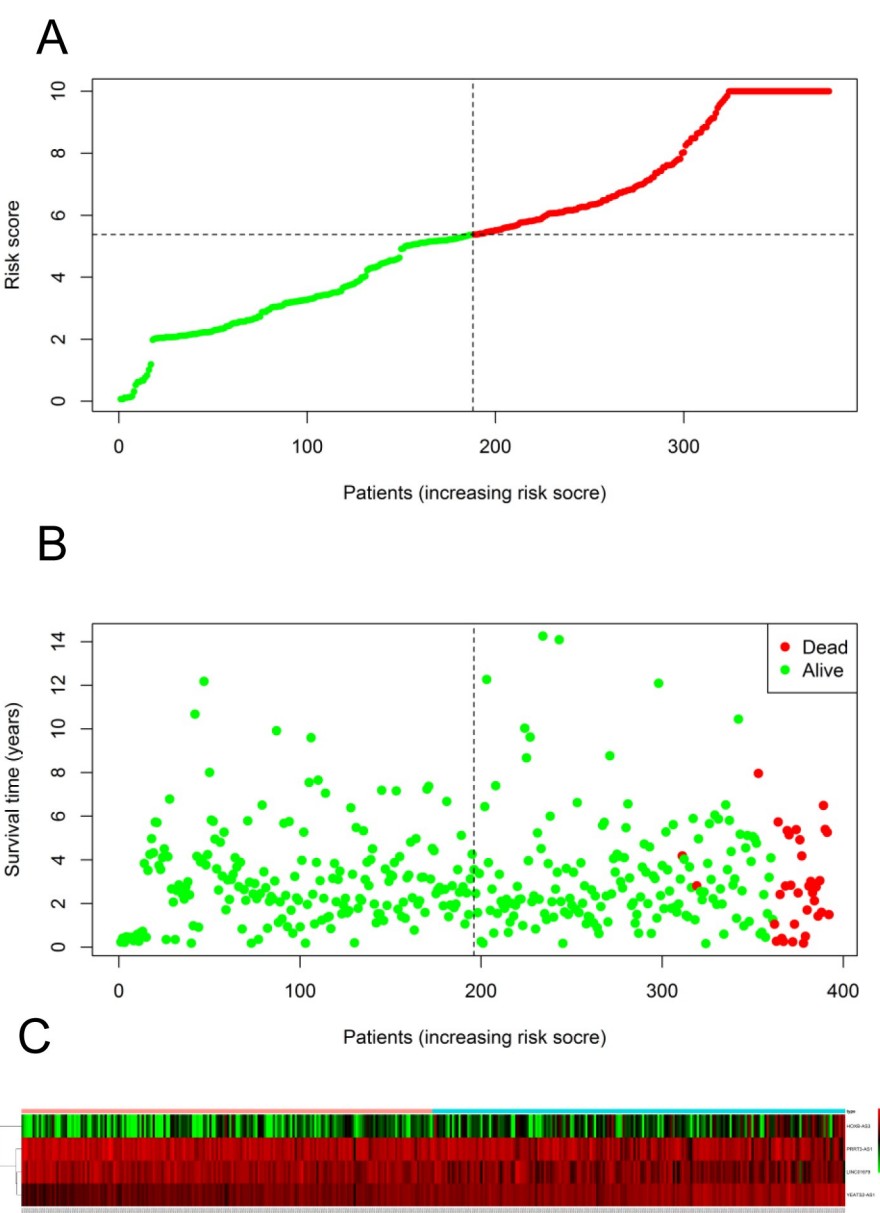

**Fig 5. Predictive power of the model for prognosis in PCa patients.** (A) Distribution of patient risk score. (B) Survival time of PCa patients.(C) Heat maps of the four lncRNAs included in the prognostic model. The vertical dotted black line represents the optimal threshold for classifying cases into high—and low-risk groups.

located in the extracellular matrix (ECM) and cell membrane, and their functions are mostly associated with the adhesion, activation and transport of substances across the extracellular matrix and cell membrane(such as GO: 0003779: actin binding, GO: 0022803: passive trans-membrane transporter activity, GO: 0015267: channel activity, GO: 0005216: ion channel activity, GO: 0005178: integrin binding, GO: 0005539: glycosaminoglycan binding, GO: 0022836: gated channel activity, GO: 0019955: cytokine binding, GO: 0098631: cell adhesion mediator activity, GO: 0046873: metal ion transmembrane transporter activity) (**Fig 16A** and **S2 Fig** and **S5 Table**). Three KEGG pathways were enriched which are mainly concerned with the interaction and binding between cells or between cells and ECM, and cell proliferation

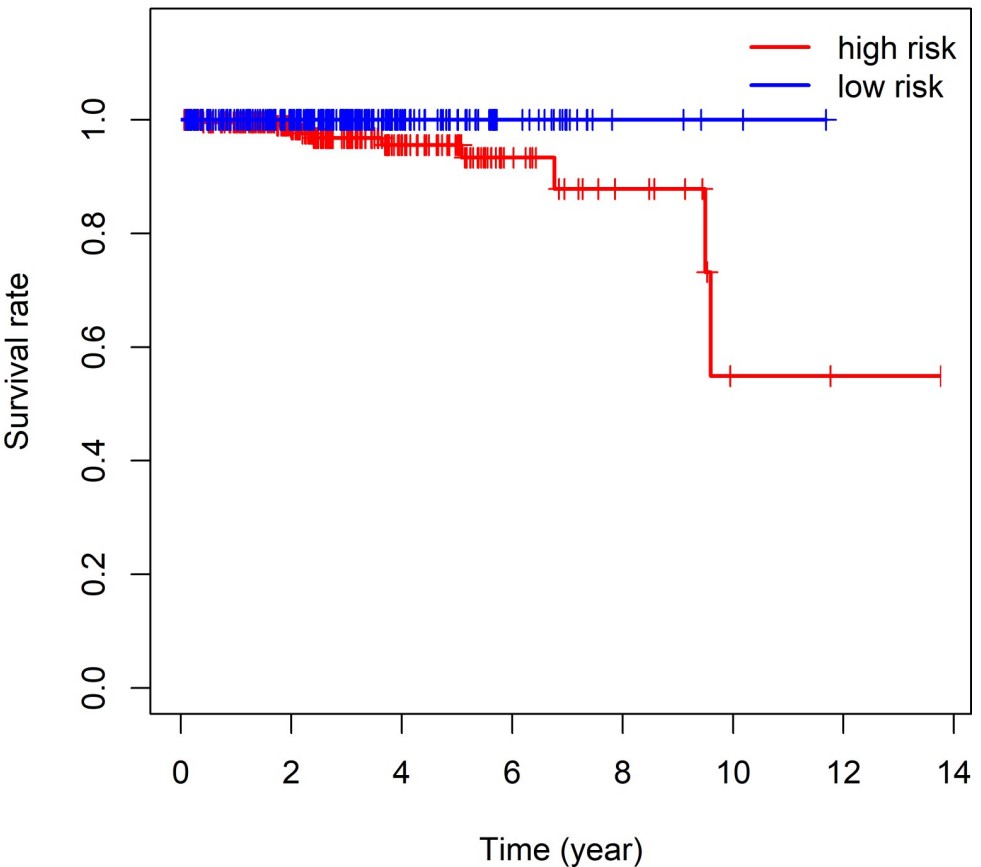

**Fig 6. Prediction ability of the four-lncRNA nomogram evaluated by Kaplan-Meier analysis, log rank P and C-index based on TCGA data.** OS time between low- and high-risk groups were visualized and compared by plotting the Kaplan-Meier curves.

(including hsa04390: Hippo signaling pathway, hsa04514: Cell adhesion molecules (CAMs), hsa04510: Focal adhesion) (**Fig 16B** and **S3, S4 Figs** and **S6 Table**).

## Discussion

Cancer risk assessment tools are essential for individualized clinical diagnosis and personalized treatment. However, the traditional clinicopathological factors, risk stratification among PCa cases represented by Gleason Score face great challenges [8]. Therefore, it needs to be addressed to establish more sensitive and effective prediction model for PCa as soon as possible. Accumulating lncRNA research publications has provided us with a new perspective and avenue for diagnosing and treating diseases like cancers [4]. Evidence is mounting that a variety of spectrum of disease progression, including tumors, is often accompanied by aberrant expression of lncRNAs, which means it may be used to be the factor to independently predict prognosis for these diseases [9]. Hence, the present work thoroughly analyzed lncRNA profiles in PCa tissues together with the matched non-carcinoma tissues against TCGA and ICGC database. Our use of TCGA and ICGC RNA sequencing data ensured the maximum detection of lncRNAs. The 4 lncRNAs-based nomogram exhibits good discriminating ability, and the

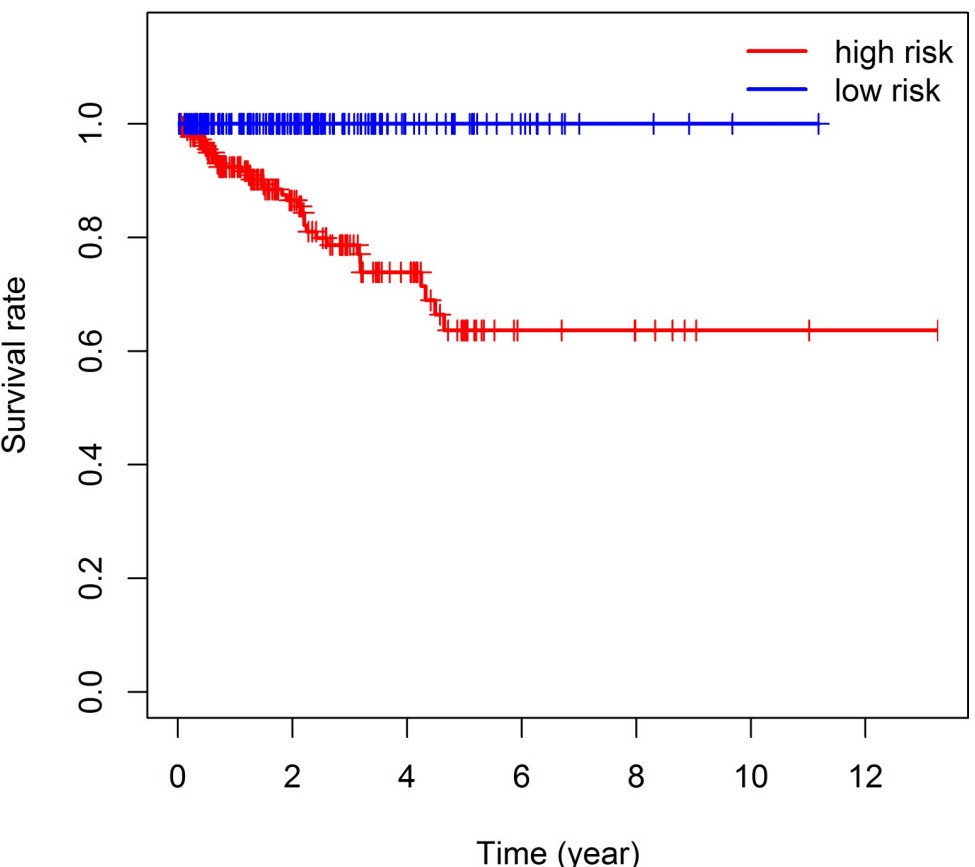

**Fig 7. Kaplan-Meier analysis, Log rank P, and C-index were used to evaluate the predictive power of four lncRNA nomogram based on ICGC data.**

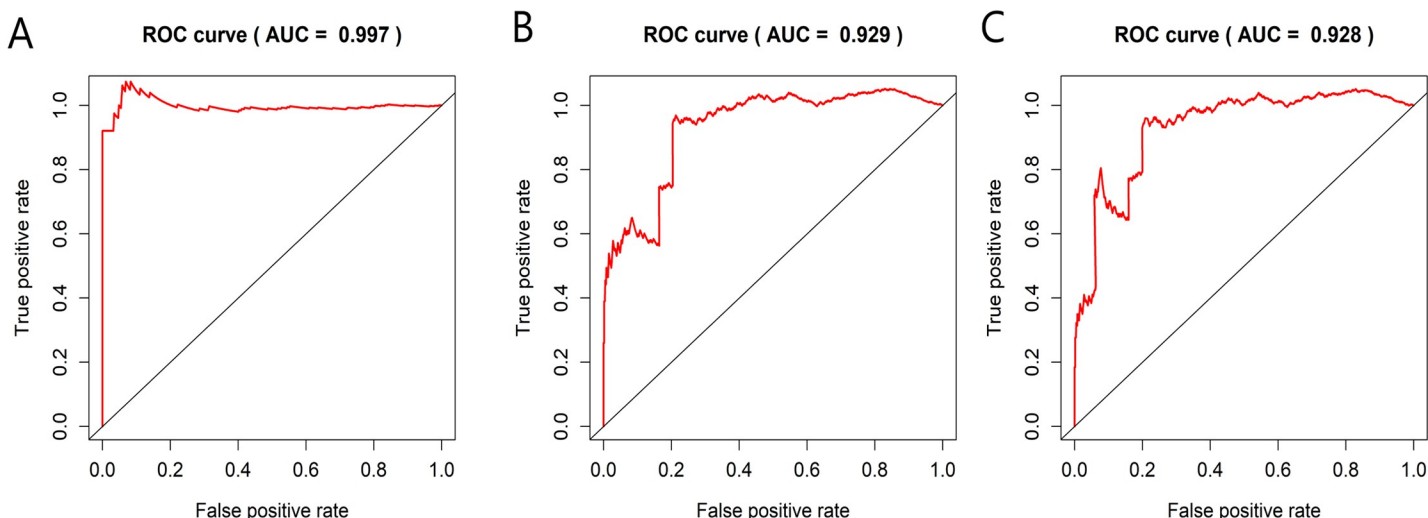

**Fig 8. Prognosis prediction ability of the four-lncRNA nomogram evaluated by ROC curves and dynamic AUC lines using TCGA data.** The time-dependent ROC curves at 1 (A), 3 (B) and 5 (C) years during the follow-up period, together with dynamic AUC lines were drawn for the cases.

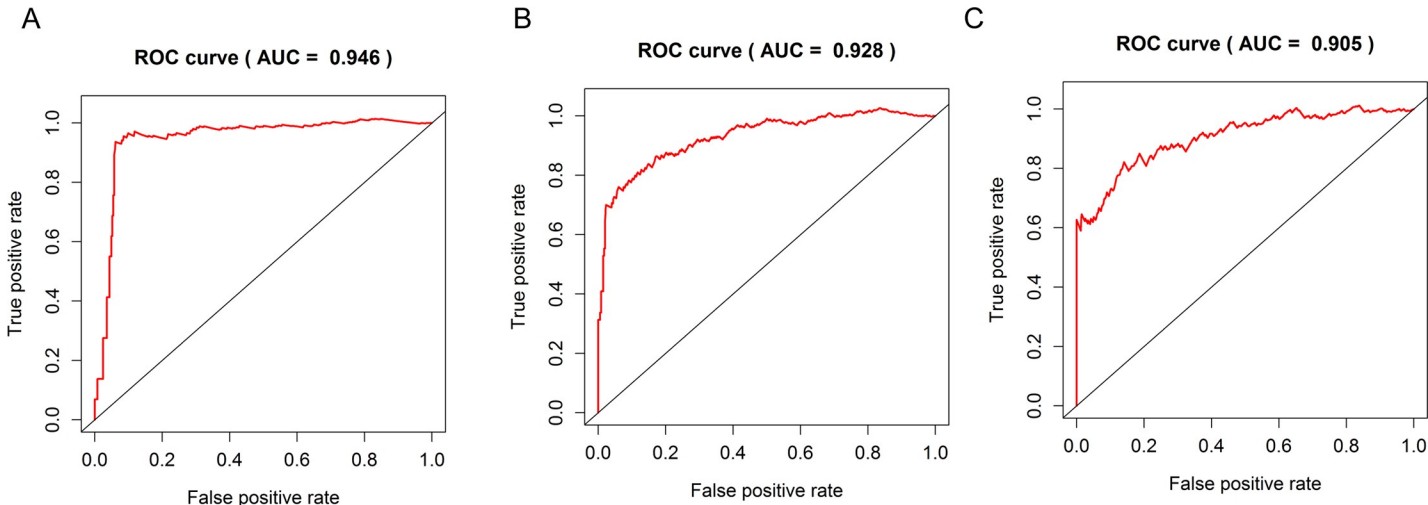

**Fig 9. Using ICGC data, the ROC curve and dynamic AUC line were used to evaluate the prognostic prediction ability of four lncRNA histograms.** The ROC curve and the area under the dynamic curve (AUC) during 1 (A), 3 (B) and 5 (C) years of follow-up were plotted.

**Table 4. TCGA OS-related univariate as well as multivariate Cox regression analysis.**

| Predictors | Univariate analysis | | | | Multivariate analysis | | | |
|---|---|---|---|---|---|---|---|---|
| | HR | HR.95L | HR.95H | P value | HR | HR.95L | HR.95H | P value |
| riskScore | 1.011634053 | 1.00650523 | 1.01678901 | 8.18E-06 | 1.013586794 | 1.004722569 | 1.022529225 | 0.002601825 |
| Age ($< 60$ Y, $\geq 60$ Y) | 1.063724213 | 0.959190535 | 1.179650091 | 0.241797732 | 1.099128551 | 0.971013403 | 1.244147165 | 0.134973704 |
| T stage (T2, T3-4) | 2.140172886 | 0.567879531 | 8.065689517 | 0.260992904 | 0.613795488 | 0.117322963 | 3.211177866 | 0.563178046 |
| N stage (N0, N1) | 3.50500237 | 0.775413395 | 15.84321563 | 0.103209029 | 1.409992903 | 0.085715392 | 23.19396706 | 0.809959377 |
| M stage (M0, M1) | 1.714435831 | 0.330971655 | 8.880791385 | 0.520628659 | 0.874796352 | 0.053104922 | 14.4105034 | 0.925446246 |
| gleasonScore ($< 8$, $\geq 8$) | 2.363916745 | 1.084202815 | 5.15411167 | 0.030522261 | 2.623777794 | 0.897852213 | 7.667419884 | 0.077895527 |

*HR-hazard ratio

**Table 5. ICGC OS-related univariate as well as multivariate Cox regression analysis.**

| Predictors | Univariate analysis | | | | Multivariate analysis | | | |
|---|---|---|---|---|---|---|---|---|
| | HR | HR.95L | HR.95H | P value | HR | HR.95L | HR.95H | P value |
| riskScore | 1.008885 | 1.005802 | 1.011978 | 1.47E-08 | 1.011292 | 1.006408 | 1.016199 | 5.46E-06 |
| Age ($< 60$ Y, $\geq 60$ Y) | 1.05216 | 0.995921 | 1.111574 | 0.06966 | 1.087683 | 1.01382 | 1.166927 | 0.019156 |
| T stage (T2, T3-4) | 2.485013 | 1.325227 | 4.659797 | 0.004542 | 0.654437 | 0.314732 | 1.360799 | 0.256314 |
| N stage (N0, N1) | 4.236203 | 2.119618 | 8.466345 | 4.38E-05 | 1.048254 | 0.40362 | 2.722456 | 0.922903 |
| M stage (M0, M1) | 6.141894 | 3.072219 | 12.2787 | 2.81E-07 | 3.839962 | 1.363908 | 10.81107 | 0.010846 |
| gleasonScore ($< 8$, $\geq 8$) | 2.85365 | 1.887136 | 4.315174 | 6.70E-07 | 2.044733 | 1.23969 | 3.372562 | 0.005086 |

*HR-hazard ratio

AUC values at 1, 3 and 5 years were 0.997, 0.929 and 0.928, separately for the whole TCGA dataset. The AUC of ICGC dataset at year 1, 3 and 5 were 0.946, 0.928 and 0.905, respectively. The as-constructed lncRNAs-based signature in this study can serve as a novel tool and shed more lights on PCa diagnosis and treatment. The high-risk PCa cases may undergo systemic

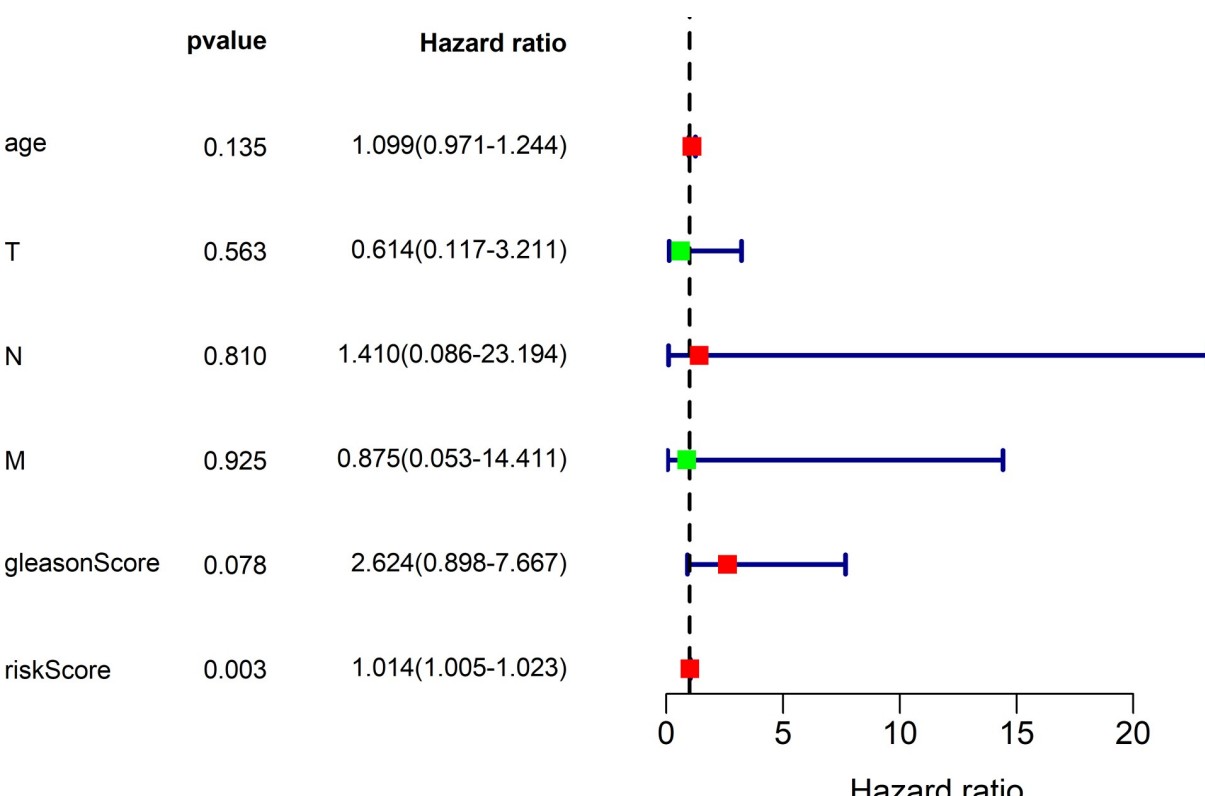

**Fig 10. TCGA multivariate analysis results demonstrated the possibility to use the 4-lncRNA signature as the potent predicting factor for PCa OS rate from other clinicopathological factors.** Red represents high risk indicators and green represents low risk ones.

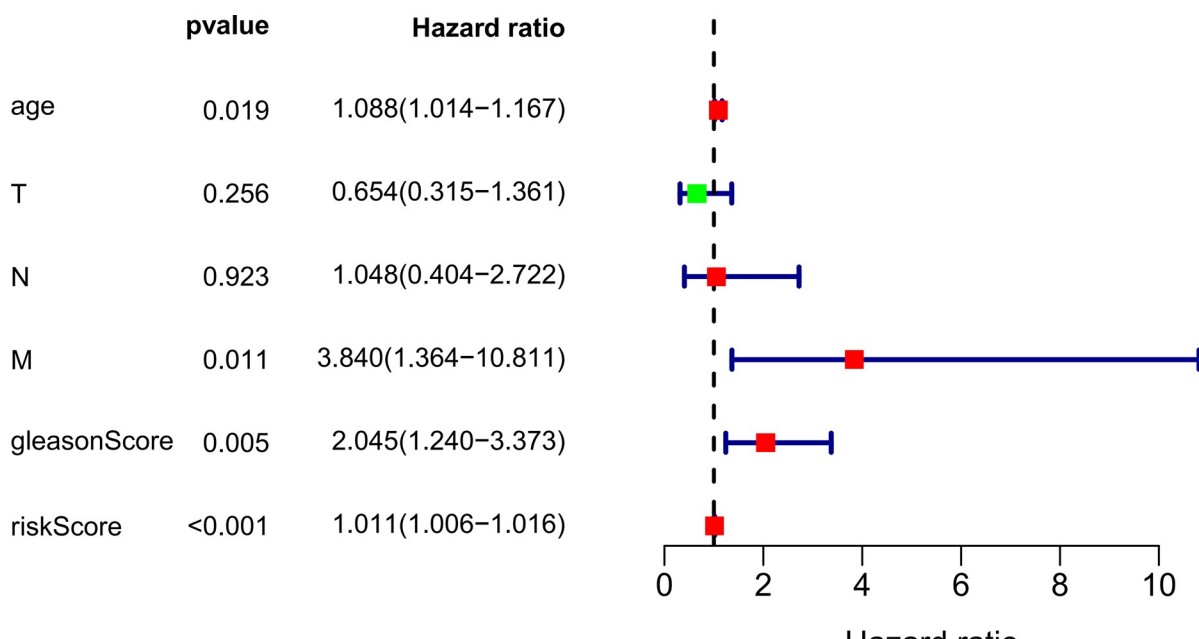

**Fig 11. The results of ICGC multivariate analysis showed that 4-lncRNA characteristics could be used as an effective predictor of PCa OS rate for other clinicopathological factors.**

## Survival curve (p=2.359e-02)

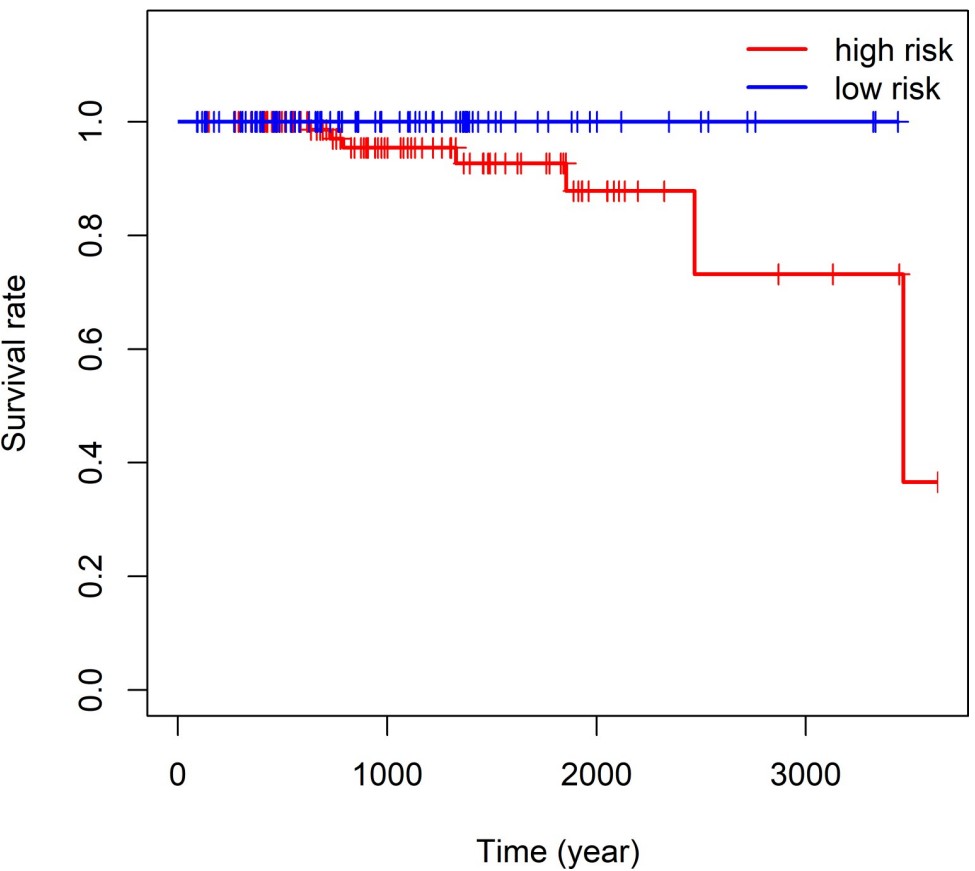

**Fig 12. TCGA Kaplan-Meier survival analysis for PCa cases having the Gleason score $> 7$.** All cases were classified as high- or low-risk group using the four-lncRNAs model.

or adjuvant therapy, and the low-risk ones may receive active surveillance. As a result, treatments can be evaluated more effectively. The four-lncRNA nomogram (HOXB-AS3, LINC01679, PRRT3-AS1 and YEATS2-AS1) was shown to be associated with patient prognosis and risk stratification upon univariate as well as multivariate COX analysis that incorporated the common clinicopathological risk factors such as age, TNM classification and Gleason score for PCa.

Gleason score represents a potent marker for evaluating PCa patient prognosis. Our multivariate COX analysis also indicated that only Gleason score and our model were good enough to predict the Overall survival risk of PCa. For PCa, Gleason score has always been adopted as an important criterion for adjuvant chemoradiotherapy or other treatments. Our study demonstrates that the four lncRNAs signature is more capable of being considered as reference factor for adjuvant therapy. Patients who had a high Gleason score may have the aggressive PCas with dismal prognosis [10]. However, Gleason score could not sufficiently accurately divide the survival risk and stage and predict prognosis of patients as a separate diagnostic indicator. Moreover, not all high-grade PCas (Gleason score $\geq 8$) are at high risk and require further adjuvant therapy [11]. As revealed by the stratification analysis, the as-constructed model assisted in classifying PCa cases with Gleason score $\geq 8$ to high- or low-risk group. Besides,

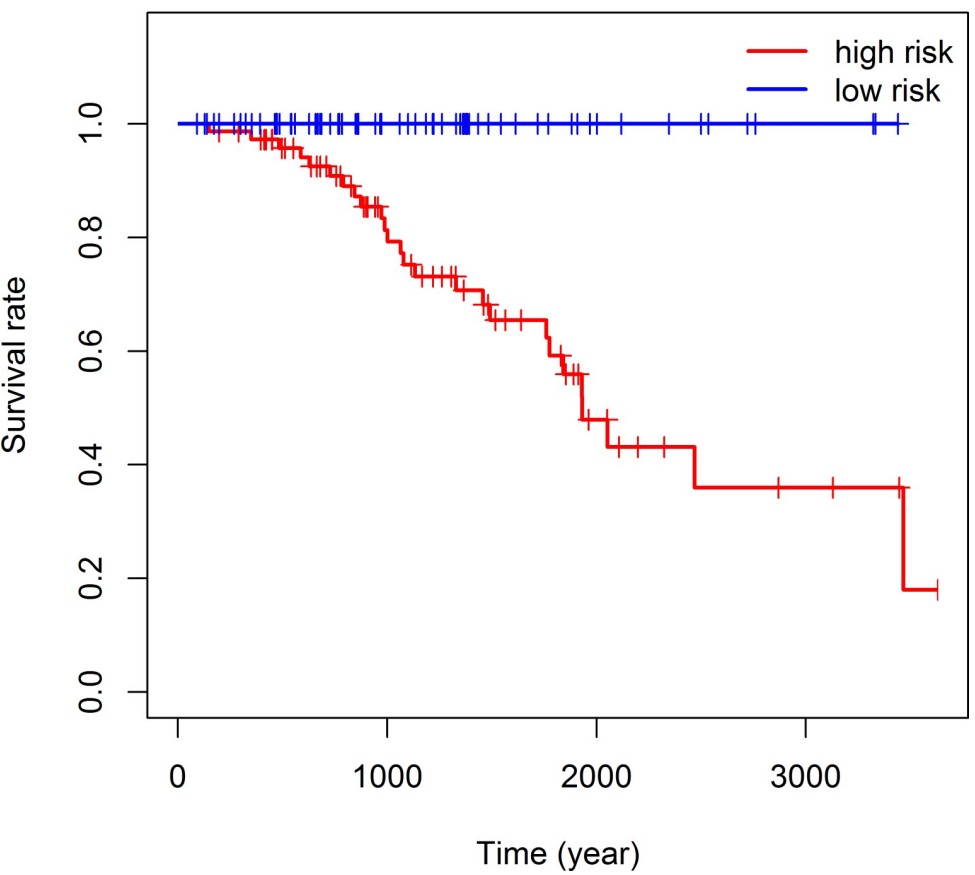

**Fig 13. ICGC Kaplan-Meier survival analysis of PCa patients with Gleason score > 7.**

our 4 lncRNAs-based model better increased the distinguishing power of Gleason score, which indicated that our constructed model helped to enhance the prediction accuracy for PCa survival risk.

Several previous studies identify lncRNAs as the excellent predictors for PCa survival. However, they were all confined to PCa biochemical recurrence patients [12–14]. In addition, in agreement with our data, Li Fan et al. found that silence of PRRT3-AS1 suppresses the proliferation of PCa cells and boost their autophagy and apoptosis [15]. It is also suggested previously that, HOXB-AS3 shows tight association with the dismal prognosis for numerous cancer types [16–18]. The above findings suggest that, HOXB-AS3 and PRRT3-AS1 can be used to be the prognostic biomarkers for numerous cancer types. Therefore, the above studies have revealed the nomogram reasonability and reliability. Moreover, such lncRNAs show possible significance in molecular targeted treatments. However, so far, little research is carried out on LINC01679 and YEATS2-AS1. Future studies should concentrate on such lncRNAs and examine the functions within PCa. Moreover, such lncRNAs possibly display possible significance for molecular targeted therapy. A majority of lncRNAs have not been functionally annotated within PCa so far, we performed the biological function clustering and associated biological signaling pathway analysis on all four lncRNAs.

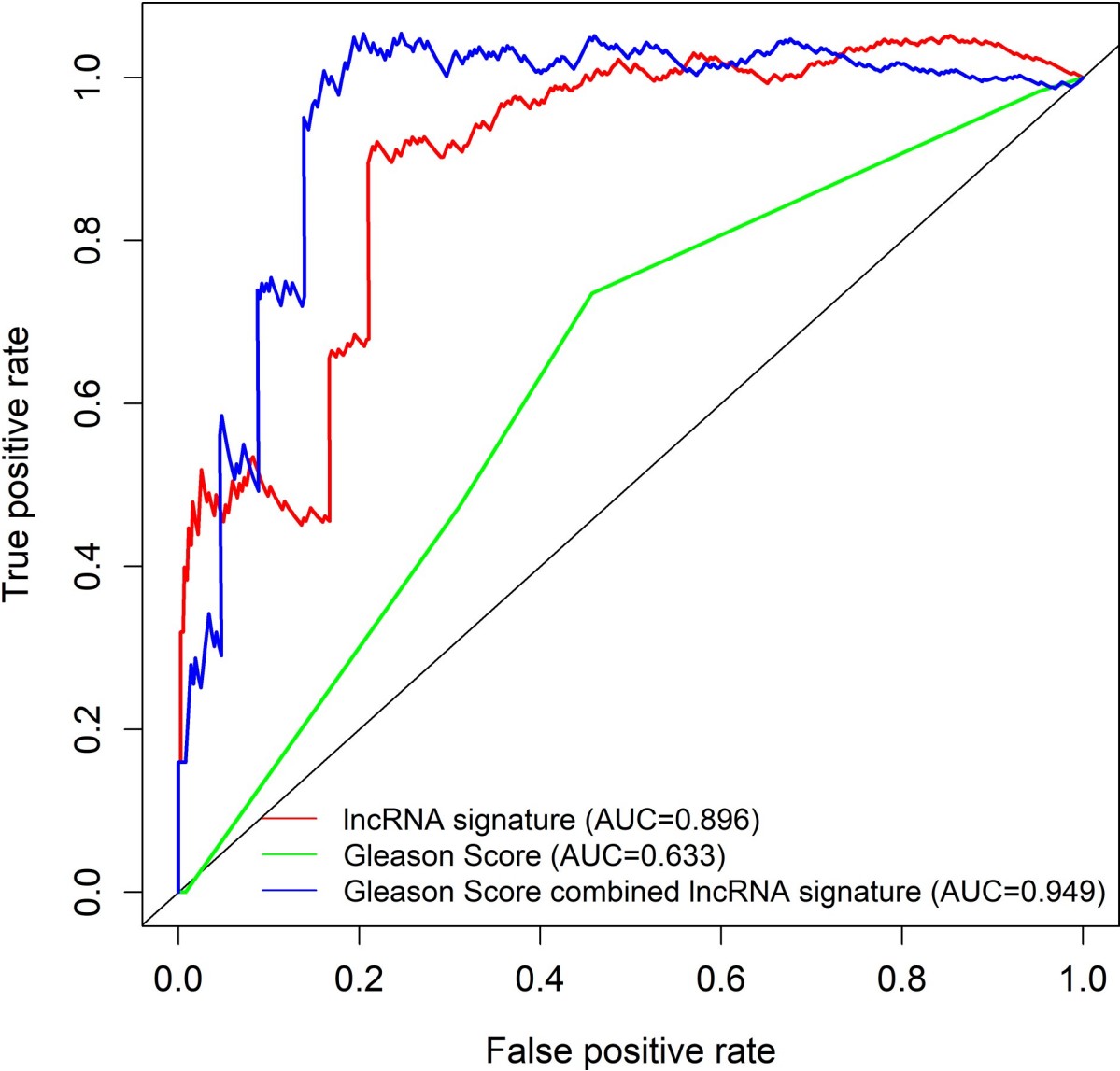

**Fig 14. TCGA ROC curves for the prediction of the 3 years overall survival among the 4-lncRNA signature model, Gleason score and our lncRNA nomogram combined with Gleason score.**

Accumulating evidence shows that lncRNAs participate in a wide range of biological processes through modulating mRNA levels epigenetically, transcriptionally, and post-transcriptionally. Nonetheless, there are still many functional annotation of lncRNAs in PCa to be further explored. This work suggested that the related biological pathway and function enrichment of those four lncRNAs by GO and KEGG. The main functional sites of our lncRNA signature are extracellular matrix and membrane, which are associated with the adhesion, activation, and transport of cellular active substances across the extracellular matrix or cell membrane. The possible molecular function analysis can shed light on future study to examine PCa incidence and development mechanisms.

Our study used genetic and clinical data from two databases, TCGA and ICGC, which covered a wide range of fields and provided a solid database for our research. At the same time, several limitations need to be acknowledged in the current study. First, the nature of

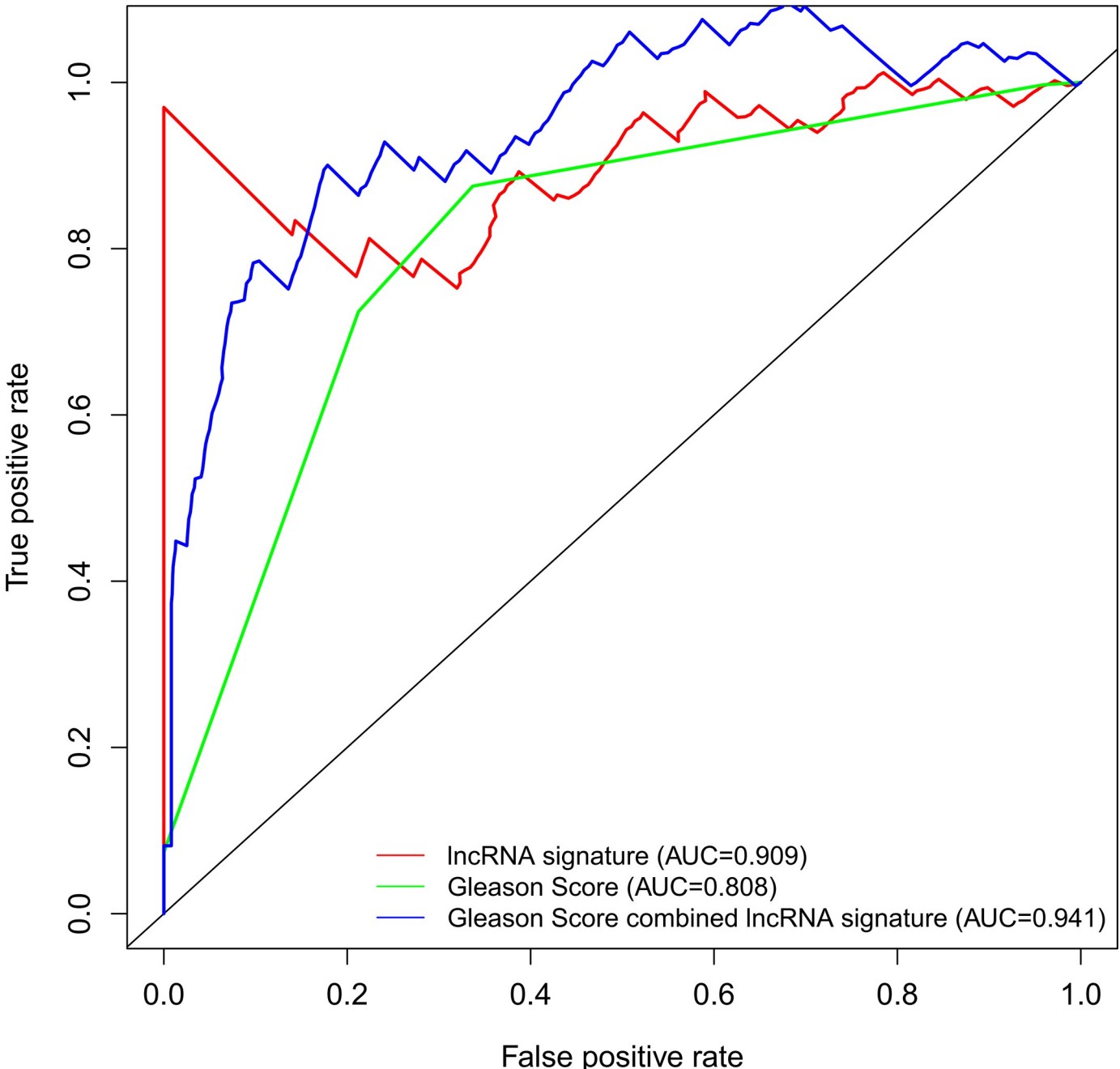

**Fig 15. ICGC ROC curves for the prediction of the 3 years Overall survival among the 4-lncRNA signature model, Gleason score and our lncRNA nomogram combined with Gleason score.**

retrospective study, such as inadequate data, is inevitable. Secondly, our analysis was based on public data without experimental verification.

## Conclusion

To sum up, some DElncRNAs between PCa and non-carcinoma tissue samples are identified in this study. Then, the 4 lncRNAs–based signature is constructed to predict the OS for PCa

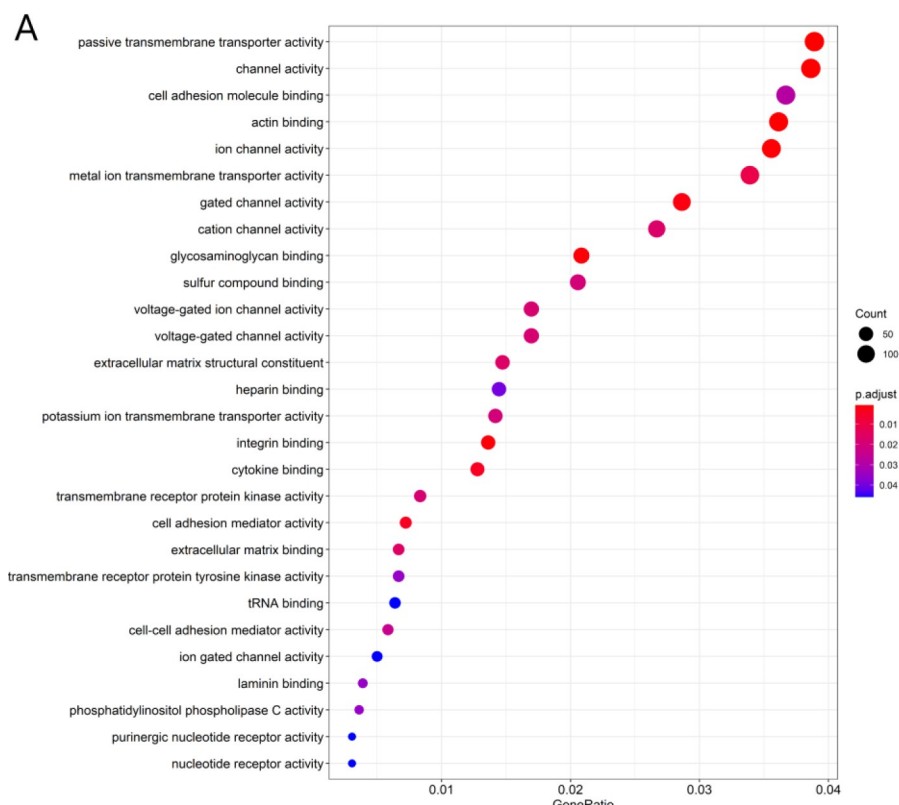

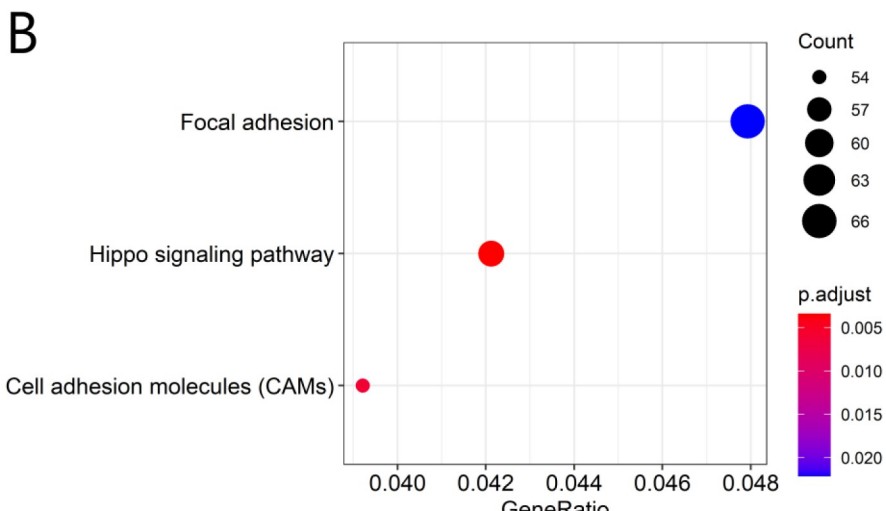

**Fig 16. Functional enrichment analysis for the related biological processes and pathways related to the 4 lncRNAs used in that model.** GO biological process enrichment results (A). KEGG signaling pathways analysis (B).

through bioinformatic analysis. As discovered by our results, our as-constructed prediction signature contributes to the effective classification of PCa as high- and low-risk groups. This prediction signature outstandingly improves the Gleason score performance in distinguishing PCa, and it can also be used to discriminate PCa (Gleason > 7). Our constructed prediction signature will assist the clinicians in taking individualized clinical treatment. However, more

studies are needed to verify our results and to examine the predicting ability of our signature for adjuvant therapy safety and efficacy. Besides, functional study is also needed to further understand molecular mechanisms underlying PCa.

## Supporting information

**S1 Fig. Unsupervised hierarchical clustering analysis of the differentially expressed lncRNAs between prostate cancer and normal tissues.**
(TIFF)

**S2 Fig. Enrichment analysis histogram of GO terms for 4-lncRNAs-related PCGs.**
(TIFF)

**S3 Fig. KEGG enrichment pathway maps for 4-lncRNAs-related PCGs. (A)** Hippo signaling pathway, **(B)** Focal adhesion and **(C)** Cell adhesion molecules (CAMs).
(ZIP)

**S4 Fig. KEGG enrichment pathways histogram for 4-lncRNAs-related PCGs.**
(TIFF)

**S1 Table. Differentially expressed lncRNAs.**
(XLSX)

**S2 Table. Univariate Cox regression result.**
(XLSX)

**S3 Table. Survival risk grouping of all prostate cancer patients from TCGA dababase.**
*futime: Survival time; *fustat: Survial state(0: Alive, 1: Dead); *riskScore: The value of risk score per patient assessed according to our 4-lncRNA model.
(XLSX)

**S4 Table. Survival risk grouping for all prostate cancer patients from the ICGC database.**
(XLSX)

**S5 Table. Enrichment analysis of GO terms for 4-lncRNAs-related PCGs.**
(XLSX)

**S6 Table. Enrichment analysis of KEGG pathways for 4-lncRNAs-related PCGs.**
(XLSX)

**S1 File.**
(RAR)

## Author Contributions

**Conceptualization:** Peng Zhang, Xuefeng Zhang.

**Investigation:** Peng Zhang, Xuefeng Zhang.

**Methodology:** Peng Zhang, Xuefeng Zhang.

**Software:** Xiaodong Tan, Daoqiang Zhang, Qi Gong.

**Writing – original draft:** Peng Zhang, Xuefeng Zhang.

**Writing – review & editing:** Peng Zhang, Xuefeng Zhang.

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
