## [Decision Letter · Decision Letter 0]

10 Dec 2020

PONE-D-20-27863

Development and validation of a set of novel and robust 4-lncRNA-based nomogram predicting prostate cancer survival by bioinformatics analysis

PLOS ONE

Dear Dr. Zhang,

Thank you for submitting your manuscript to PLOS ONE. After careful consideration, we feel that it has merit but does not fully meet PLOS ONE’s publication criteria as it currently stands. Therefore, we invite you to submit a revised version of the manuscript that addresses the points raised during the review process.

We look forward to receiving your revised manuscript.

Kind regards,

Hiromu Suzuki, M.D., Ph.D.

Academic Editor

PLOS ONE

Additional Editor Comments:

This manuscript was carefully reviewed by 2 experts, and both of them found several concerns which need to be addressed before acceptance. For instance, reviewer 1 suggested to discuss the limitation of this study. Reviewer 2 also suggested additional analysis using other datasets to validate the results. Please respond to each of the reviewer comments.

Journal Requirements:

Reviewers' comments:

Reviewer's Responses to Questions

**Comments to the Author**

1. Is the manuscript technically sound, and do the data support the conclusions?

Reviewer #1: No

Reviewer #2: Yes

2. Has the statistical analysis been performed appropriately and rigorously? 

Reviewer #1: I Don't Know

Reviewer #2: Yes

3. Have the authors made all data underlying the findings in their manuscript fully available?

Reviewer #1: Yes

Reviewer #2: Yes

4. Is the manuscript presented in an intelligible fashion and written in standard English?

Reviewer #1: Yes

Reviewer #2: Yes

5. Review Comments to the Author

Reviewer #1: The manuscript by Zhang, et al. uses the TCGA data to identify differentially expressed lncRNA between cancerous and non-cancerous tissues. In their analysis, they identified four lncRNA’s (2 protective and 2 harmful) that help improve stratification of overall survival from prostate cancer. Evaluating their data, it appears that there were only 9 deaths amongst the 397 patients who were included in the study. Therefore, since they are only evaluating overall survival, and not biochemical recurrence, distant metastasis, or progression of disease, how their results can be meaningful in making a clear utility of the four lncRNA’s.

The limitations of their study are not discussed at all. They also make overstatements that their findings can be used to determine need for adjuvant therapy. However, without validation and evaluation of other parameters, besides overall survival, they cannot make such lofty claims.

Reviewer #2: 1、The as-constructed lncRNAs-based signature in this study can serve as a novel tool and shed more lights on PCa diagnosis and treatment. The 4 lncRNAs–based signature is constructed to predict the OS for PCa through bioinformatic analysis. This prediction signature outstandingly improves the Gleason score performance in distinguishing PCa, and it can also be used to discriminate PCa (Gleason > 7).

2、It is better to add other databases analysis and related function experiments.

3、A few sentence typos，such as line78 4-lncrNA etc.

6. PLOS authors have the option to publish the peer review history of their article (what does this mean?). If published, this will include your full peer review and any attached files.

Reviewer #1: No

Reviewer #2: No

---

## [Author Response · Author response to Decision Letter 0]

12 Mar 2021

Dear Dr. Hiromu Suzuki

Thank you very much for your decision letter and advice on our manuscript (Manuscript PONE-D-20-27863) entitled “Development and validation of a set of novel and robust 4-lncRNA-based nomogram predicting prostate cancer survival by bioinformatics analysis”. We also thank the reviewers for the constructive comments and suggestions. We have revised the manuscript accordingly, and all amendments are indicated by red font in the revised manuscript. In addition, our point-by-point responses to the comments are listed below this letter.

We hope that our revised manuscript is now acceptable for publication in your journal and look forward to hearing from you soon. 

With best wishes,

Yours sincerely,

Xuefeng Zhang

 

First of all, we would like to express our sincere gratitude to the reviewers for their constructive and positive comments.

Replies to Reviewer 1

Reviewer #1: The manuscript by Zhang, et al. uses the TCGA data to identify differentially expressed lncRNA between cancerous and non-cancerous tissues. In their analysis, they identified four lncRNA’s (2 protective and 2 harmful) that help improve stratification of overall survival from prostate cancer. Evaluating their data, it appears that there were only 9 deaths amongst the 397 patients who were included in the study. Therefore, since they are only evaluating overall survival, and not biochemical recurrence, distant metastasis, or progression of disease, how their results can be meaningful in making a clear utility of the four lncRNA’s.

The limitations of their study are not discussed at all. They also make overstatements that their findings can be used to determine need for adjuvant therapy. However, without validation and evaluation of other parameters, besides overall survival, they cannot make such lofty claims.

Response: We added gene sequencing and clinical data from ICGC database to conduct additional analysis and verification of the prediction performance of our prediction model. Because the mortality rate for prostate cancer in the United States is so low, only nine of the 397 Americans from the TCGA database previously included in the study died. Because this study evaluated overall survival, we added 378 patients from the ICGC database from China, 34 of whom died. Thus, the reliability of this study is further increased. We have added a discussion of the limitations of this study.

Replies to Reviewer 2

Reviewer #2: 1、The as-constructed lncRNAs-based signature in this study can serve as a novel tool and shed more lights on PCa diagnosis and treatment. The 4 lncRNAs–based signature is constructed to predict the OS for PCa through bioinformatic analysis. This prediction signature outstandingly improves the Gleason score performance in distinguishing PCa, and it can also be used to discriminate PCa (Gleason > 7).

2、It is better to add other databases analysis and related function experiments.

3、A few sentence typos，such as line78 4-lncrNA etc.

Response: 

1. We added gene sequencing and clinical data from ICGC database to conduct additional analysis and verification of the prediction performance of our prediction model. 

2. In view of the expression errors in our paper you proposed, we have also made corresponding modifications.

3. Correction has been made in the revised manuscript

---

## [Decision Letter · Decision Letter 1]

29 Mar 2021

Development and validation of a set of novel and robust 4-lncRNA-based nomogram predicting prostate cancer survival by bioinformatics analysis

PONE-D-20-27863R1

Dear Dr. Zhang,

We’re pleased to inform you that your manuscript has been judged scientifically suitable for publication and will be formally accepted for publication once it meets all outstanding technical requirements.

Kind regards,

Hiromu Suzuki, M.D., Ph.D.

Academic Editor

PLOS ONE

Additional Editor Comments (optional):

Reviewers' comments:

Reviewer's Responses to Questions

**Comments to the Author**

1. If the authors have adequately addressed your comments raised in a previous round of review and you feel that this manuscript is now acceptable for publication, you may indicate that here to bypass the “Comments to the Author” section, enter your conflict of interest statement in the “Confidential to Editor” section, and submit your "Accept" recommendation.

Reviewer #1: All comments have been addressed

Reviewer #2: (No Response)

2. Is the manuscript technically sound, and do the data support the conclusions?

Reviewer #1: Yes

Reviewer #2: (No Response)

3. Has the statistical analysis been performed appropriately and rigorously? 

Reviewer #1: Yes

Reviewer #2: (No Response)

4. Have the authors made all data underlying the findings in their manuscript fully available?

Reviewer #1: Yes

Reviewer #2: (No Response)

5. Is the manuscript presented in an intelligible fashion and written in standard English?

Reviewer #1: Yes

Reviewer #2: (No Response)

6. Review Comments to the Author

Reviewer #1: (No Response)

Reviewer #2: (No Response)

7. PLOS authors have the option to publish the peer review history of their article (what does this mean?). If published, this will include your full peer review and any attached files.

Reviewer #1: No

Reviewer #2: No

---

## [Editor Report · Acceptance letter]

19 Apr 2021

PONE-D-20-27863R1 

Development and validation of a set of novel and robust 4-lncRNA-based nomogram predicting prostate cancer survival by bioinformatics analysis 

Dear Dr. Zhang:

I'm pleased to inform you that your manuscript has been deemed suitable for publication in PLOS ONE. Congratulations! Your manuscript is now with our production department. 

Kind regards, 

on behalf of

Dr. Hiromu Suzuki 

Academic Editor

PLOS ONE